# Prediction of Multi-Site PM$_{2.5}$ Concentrations in Beijing Using CNN-Bi LSTM with CBAM

**Dong Li** [1,2,3,4], **Jiping Liu** [2] **and Yangyang Zhao** [2,*]

1    Faculty of Geomatics, Lanzhou Jiaotong University, Lanzhou 730070, China
2    Chinese Academy of Surveying and Mapping, Beijing 100830, China
3    National-Local Joint Engineering Research Center of Technologies and Applications for National Geographic State Monitoring, Lanzhou 730070, China
4    Gansu Provincial Engineering Laboratory for National Geographic State Monitoring, Lanzhou 730070, China
*    Correspondence: zhaoyy@casm.ac.cn

**Abstract:** Air pollution is a growing problem and poses a challenge to people's healthy lives. Accurate prediction of air pollutant concentrations is considered the key to air pollution warning and management. In this paper, a novel PM$_{2.5}$ concentration prediction model, CBAM-CNN-Bi LSTM, is constructed by deep learning techniques based on the principles related to spatial big data. This model consists of the convolutional block attention module (CBAM), the convolutional neural network (CNN), and the bi-directional long short-term memory neural network (Bi LSTM). CBAM is applied to the extraction of feature relationships between pollutant data and meteorological data and assists in deeply obtaining the spatial distribution characteristics of PM$_{2.5}$ concentrations. As the output layer, Bi LSTM obtains the variation pattern of PM$_{2.5}$ concentrations from spatial data, overcomes the problem of long-term dependence on PM$_{2.5}$ concentrations, and achieves the task of accurately forecasting PM$_{2.5}$ concentrations at multiple sites. Based on real datasets, we perform an experimental evaluation and the results show that, in comparison to other models, CBAM-CNN-Bi LSTM improves the accuracy of PM$_{2.5}$ concentration prediction. For the prediction tasks from 1 to 12 h, our proposed prediction model performs well. For the 13 to 48 h prediction task, the CBAM-CNN-Bi LSTM also achieves satisfactory results.

**Keywords:** air pollution; PM$_{2.5}$ concentrations; multiple sites; CBAM; CNN; Bi LSTM

## 1. Introduction

Over the past few years, significant social concern has been generated by the increasingly serious issue of air pollution. [1]. Prediction of pollutant concentrations in the air plays an essential role in environmental management and air pollution prevention [2]. PM$_{2.5}$ (particulate matter less than 2.5 µm in diameter) is an important component of air pollutants [3]. Therefore, the prediction of PM$_{2.5}$ concentration trends is considered a critical issue in the prediction of air pollutant concentrations.

Deterministic and statistical approaches can be used to forecast PM$_{2.5}$ concentrations based on the features of the study methods. [4]. Deterministic methods simulate the emission, dispersion, transformation, and removal of PM$_{2.5}$ through meteorological principles and statistical methods [5], thus enabling the prediction of PM$_{2.5}$ concentrations. There are several representative models for pollutant concentration prediction based on deterministic methods. There are a few representative models for forecasting PM$_{2.5}$ concentrations based on deterministic methods: a Community Multiscale Air Quality Modeling System (CMAQ) [6], a nested air quality prediction modeling system (NAQPMS) [7], and a Weather Research and Forecasting Model with Chemistry (WRF-Chem) [8].

Different from the deterministic methods, the statistical methods do not have complex theoretical models, and they give better predictions through the learning and analysis of historical data on pollutants. Statistical methods are mainly classified into two approaches:

machine learning approaches and deep learning approaches [9]. The classical machine learning models used for the prediction of $PM_{2.5}$ concentrations include Random Forest (RF) [10] models, Autoregressive Sliding Average (ARMA) Models [11], Autoregressive Integrated Moving Average (ARIMA) Models [12], Support Vector Regression (SVR) [13], and Linear Regression (LR) models [14].

Compared with machine learning methods, which suffer from slow convergence and inadequate generalization [5], deep learning is widely used in $PM_{2.5}$ concentration prediction due to its ability to fit data more robustly and non-linearly [15]. The following deep learning models have been used to forecast pollution concentrations: Convolutional neural networks (CNN) [16], Back Propagation Neural Networks (BPNN) [17], Recurrent Neural Networks (RNN) [18], Gate Recurrent Units (GRU) [19], Long Short-Term Memory networks (LSTM) [20], and Bi-directional Long Short-Term Memory Neural Networks (Bi LSTM) [21], attention-based ConvLSTM (Att-ConvLSTM) [22], etc. Although the above models are widely used in $PM_{2.5}$ concentration prediction due to their superiority in handling time series data, the current pollutant concentration prediction models described above have the following problem: owing to the single network model, it is limited by the feature dimension of the input data, in other words, the dimension of the hidden state is influenced by the dimension of the input data.

In order to solve the problem that the predictive power of a single deep learning network is limited, in recent years, hybrid deep learning models have been widely used in the research of pollutant concentration prediction. Hybrid deep learning models have several different network structures that allow better quantification of complex data [23], which have been used for pollutant concentration prediction, including: LSTM-FC [24], AC-LSTM [25], EEMD-GRNN [26], etc. Meanwhile, air pollution is a problem of regional dispersion with spatial dimensions [27], and there is a spatial interrelationship of air pollution between adjacent sites. However, all of the above hybrid deep learning models have focused on the prediction of pollutant concentrations at individual stations and have not considered the spatial correlation of adjacent observation sites. CNN-LSTM [15] is a recently proposed hybrid deep learning model, which can handle time series problems and has successfully interpreted the spatial distribution characteristics of air pollutant concentrations through the image analysis capability of CNN [15]. However, there are also three crucial problems with CNN-LSTM [28]. Firstly, the simple structure of CNN leads to the loss of feature information and the inability to extract deep spatial features of contaminant data [29]. Secondly, CNN-LSTM has difficulty capturing the long time-series variation between pollutant concentrations. Finally, much of the work at the present stage confirms the complex interactions between pollutant data and meteorological data [5]. CNN-LSTM has trouble obtaining complicated correlation characteristics between meteorological input and air pollution input. In view of the above, we introduced the convolutional block attention module (CBAM) to build a CNN-based prediction model: the CBAM-CNN-Bi LSTM model. The reasons are as follows.

(1) The convolutional block attention module includes the channel attention module (CAM) and the spatial attention module (SAM) [30]. It is a simple and effective attention module that can be arbitrarily embedded into any 2D CNN model and does not consume too much of the computer's running memory.

(2) As the network depth is increased, convolutional neural networks degrade and lose feature information. Therefore, we introduce a spatial attention module to efficiently extract spatially relevant features of contaminant data between multiple stations [29].

(3) The correlation characteristics between pollution data and meteorological data are not taken into account by the aforementioned prediction models. In order to optimize the prediction outcomes based on the intricate correlation characteristics of the model input data, we thoroughly analyze the prediction problems of pollution data and meteorological data at each station and present the channel attention module.

In this study, our proposed prediction model is fully taken into account to produce more accurate predictions of $PM_{2.5}$ concentrations in the target city in the future, which

should achieve the following goals: (1) efficiently utilizing historical pollutant concentration and meteorological big data from multiple stations; (2) accurately achieving long-term predictions of pollutant concentrations in the target city; and (3) in-depth exploration of the spatial and temporal correlation characteristics from multiple stations.

The remainder of the essay is structured as follows. The study area, the experimental data and the procedures for processing the experimental data, as well as the overall framework of the pollutant concentration prediction model, and a detailed explanation of each model component are described in Section 2. The key findings and expectations are covered in Section 3. The work is concluded in Section 4, which also suggests areas for future investigation.

## 2. Materials and Methods

### 2.1. Materials

#### 2.1.1. Data

The experimental information in this study consists of pollutant data and meteorological data. The data was monitored by twelve air quality monitoring stations in the Beijing area from 1 March 2013 to 28 February 2017. The distribution of the twelve stations is shown in Figure 1a. The meteorological data and pollutant data selected are from the UCI website (https://archive.ics.uci.edu/ml/datasets/Beijing+Multi-Site+Air-Quality+Data, accessed on 1 April 2022). Figure 1b shows a time series plot of $PM_{2.5}$ concentration data from twelve stations for the period 1–31 December 2016. The meteorological data include hourly temperature, pressure, dew point, precipitation, wind direction, and wind speed; and the pollutant data include hourly $PM_{2.5}$, $PM_{10}$, $SO_2$, $NO_2$, CO, and $O_3$. Table 1 shows the range of values for meteorological data and pollutant data.

**Table 1.** Experimental data of the $PM_{2.5}$ concentration prediction model.

| Type | Variable | Unit |
|---|---|---|
| Meteorological Data | Temperature | $[-19.9, 41.6]$ |
| | Pressure | $[982.4, 1042.8]$ |
| | Dew Point | $[-43.4, 29.1]$ |
| | Precipitation | $[0, 72.5]$ |
| | Wind Direction | $[N, ESE]$ |
| | Wind Speed | $[0, 13.2]$ |
| Pollutant Data | $PM_{2.5}$ | $[2, 999]$ |
| | $PM_{10}$ | $[2, 999]$ |
| | $SO_2$ | $[0.2856, 500]$ |
| | $NO_2$ | $[1.0265, 290]$ |
| | CO | $[100, 10000]$ |
| | $O_3$ | $[0.2142, 1071]$ |

#### 2.1.2. Data Preprocessing

Wind direction, as a non-numerical type of data, needs to be converted to a numerical type of data by categorical coding. The average of the data prior to and following the time of the missing value is used to fill in missing values for meteorological and pollution data. Then, in order to eliminate the effect of numerical differences on prediction accuracy, meteorological and pollutant data were converted to the range [0, 1] by the Min-Max function as below.

$$y' = \frac{y - \min(y)}{\max(y) - \min(y)} \quad (1)$$

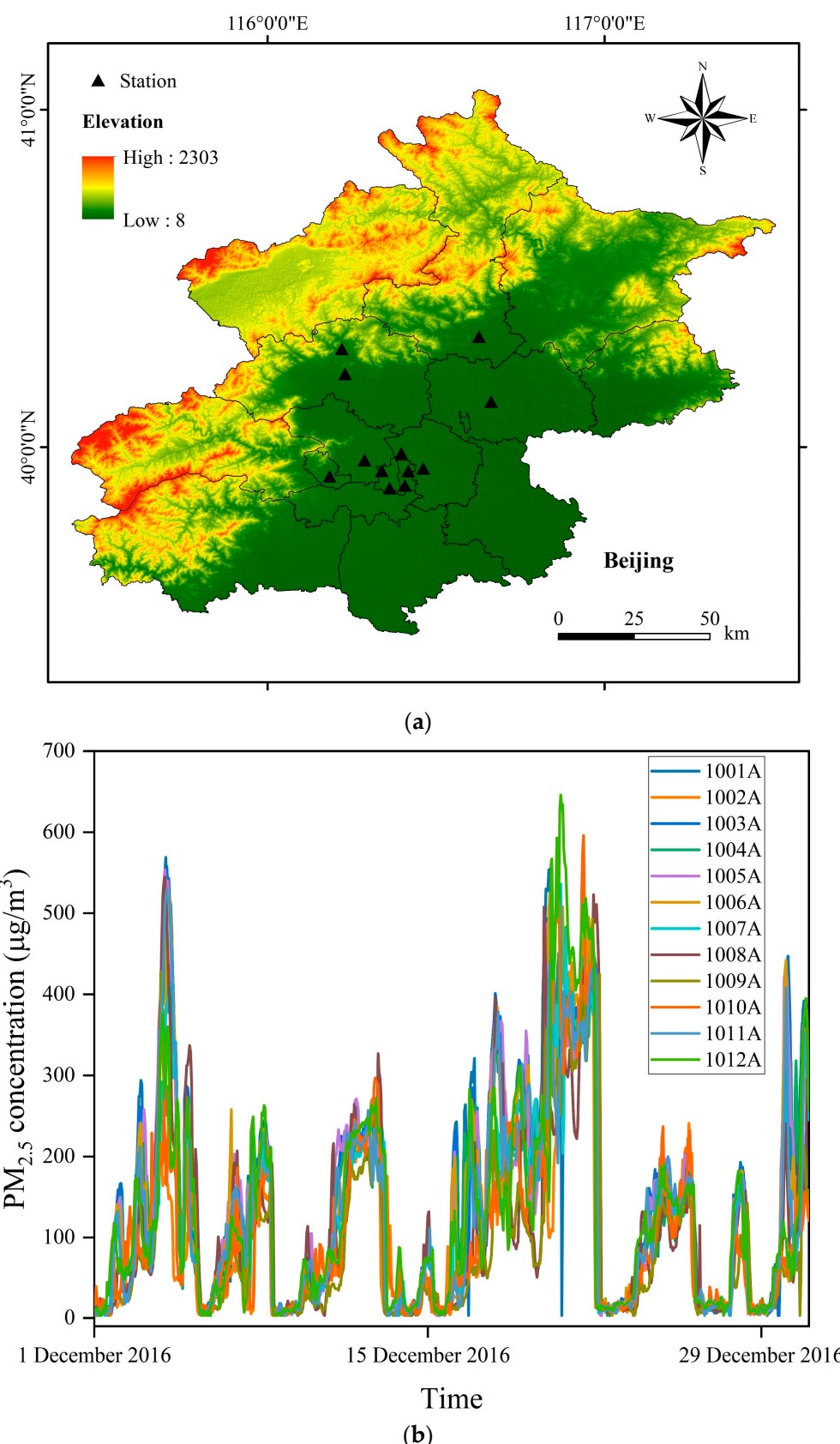

(**a**)

(**b**)

**Figure 1.** (**a**). Distribution of air quality monitoring stations in Beijing. (**b**). Time series plot of PM$_{2.5}$ concentration data.

### 2.2. Methods

### 2.2.1. CBAM-CNN-Bi LSTM

Deep learning is a type of machine learning that can train samples of data using unsupervised approaches in order to create deep network structures [31]. We propose a CBAM-CNN-Bi LSTM model, whose intricate structure is depicted in Figure 2, to reliably anticipate $PM_{2.5}$ concentrations. First, we took advantage of CNN properties to identify key characteristics in the input $PM_{2.5}$ data and to obtain the spatial dependence of all $PM_{2.5}$ sites. Then, we capture the time dependence of the $PM_{2.5}$ series data using the special architecture of Bi LSTM for time-series problems. In addition, we added the convolutional block attention module to the CNN to enhance training accuracy by focusing on channel and spatial information. The convolutional block attention module is classified into the channel attention module and the spatial attention module. The channel attention module enables the network to disregard the remainder and concentrate on the useful feature channels. The spatial attention module enables the network to concentrate on the nearby areas on the feature map [32]. In other words, the channel attention module enables the network to focus on the classes of factors which have a greater impact on $PM_{2.5}$ concentrations, and the spatial attention module enables the network to focus on areas where there is a stronger spatial relationship between $PM_{2.5}$ sites.

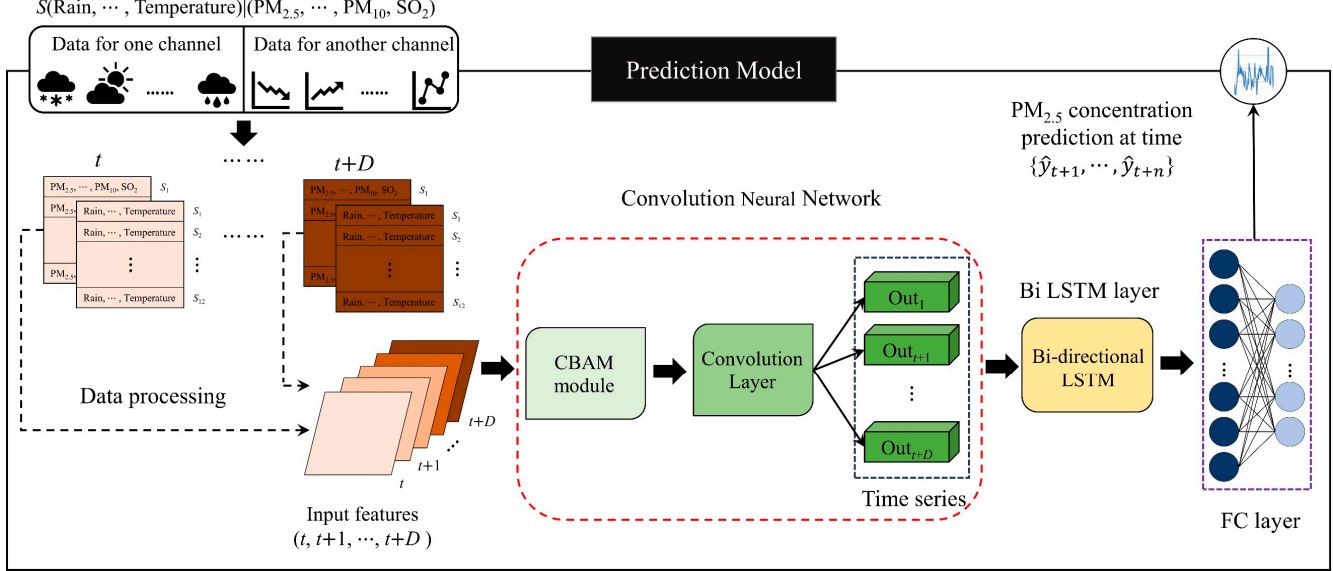

**Figure 2.** The architecture of CBAM-CNN-Bi LSTM.

The CNN serves as the basis layer for the CBAM-CNN-Bi LSTM model, and its convolutional layer is used to extract features. The convolutional block attention module is embedded in the CNN, and the channel and spatial feature information in the convolutional block attention module capture is used as an input for the CNN. For time series prediction, the higher layer Bi LSTM uses the output of the CNN layer as its input. In Figure 2, the prediction model is displayed.

Meteorological data and pollutant concentrations are converted into two-dimensional matrices with time series as inputs to the prediction model. These matrices are then fed into the convolutional block attention module and the CNN network in order to obtain characteristics. Its output serves as the Bi LSTM's input. The fully connected layer decodes the Bi LSTM's output to get the final prediction result.

### 2.2.2. Convolutional Block Attention Module

Figure 3a shows the architecture of the convolutional block attention module [33], which can perform attention in both the channel dimension and the spatial dimension

by concentrating on important elements while disregarding unimportant ones. There are two independent sub-modules in the convolutional block attention module, the channel attention module and the spatial attention module, and the structure of each of these two sub-modules is shown in Figure 3b,c. The characteristic map is created by the convolutional layer, and the weighting of the characteristic map is calculated along the order of the channel attention module first and then the spatial attention module. Then, the weight map and the input feature map are multiplied to carry out adaptive feature optimization learning [30]. The convolutional block attention module is designed as a simple attention block which is casually embedded into any 2D CNN model for end-to-end training and improved model representation at a lower cost [33].

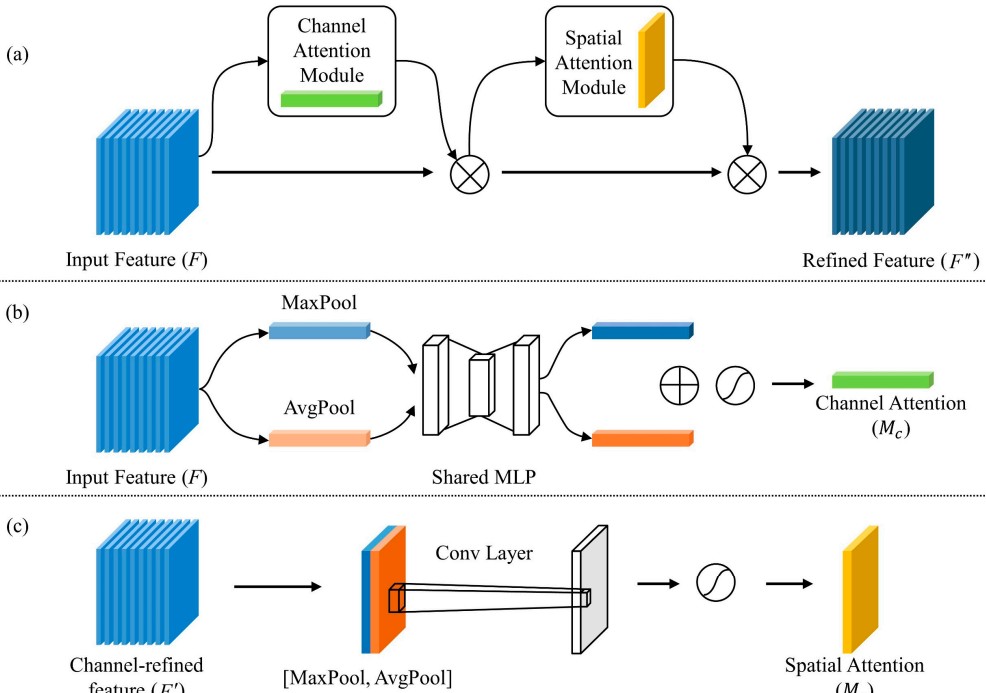

**Figure 3.** The architecture of convolutional block attention module. (**a**) The convolutional block attention module. (**b**) The architecture of channel attention. (**c**) The architecture of spatial attention. Reproduced with permission from Ref. [34]. Copyright 2021 ELSEVIER.

As shown in Figure 3a, the convolutional block attention module combines the spatial attention and channel attention modules to infer attention weight maps and produces detailed characteristic maps. These two sub-modules together are referred to as:

$$F' = M_c(F) \otimes F \tag{2}$$

$$F'' = M_s(F') \otimes F' \tag{3}$$

Here, $\otimes$ denotes element-wise multiplication, $F$ represents the input feature map, $F'$ represents the channel-refined characteristic map, $F''$ is the refined characteristic map.

The channel attention module identifies the more important channels based on their relationship to each other. The channel attention module has two pools, *Maxpool* and *Avgpool*($\cdot$). Firstly, the input features are extracted by *Maxpool* and *Avgpool*($\cdot$) for different high-level features. By using shared MLP (multi-layer perceptron), the two categories of high-level characteristics are then combined. Lastly, a sigmoid function activates the fused features to show the channel priority of the input characteristics. The channel attention module's computation is done in the manner listed below.

$$M_c(F) = f_{sigmoid}(MLP(MaxPool(F) + MLP(AvgPool(F)))) \tag{4}$$

The spatial attention module acts as a complement to the channel attention module, and it focuses on which position of information is more important. At first, *Maxpool* and *Avgpool*(·) are performed on the input features. Then, the outputs of two different features are connected to generate a novel characteristic descriptor. Lastly, the new feature descriptors are transformed into refined features through the convolution and sigmoid function operations. The following describes how the SAM is calculated.

$$M_s(F) = f_{sigmoid}(Conv[MaxPool(F); AvgPool(F)])) \tag{5}$$

where *Maxpool*(·) represents maxpooling, *Avgpool*(·) represents averagepooling, *MLP*(·) represents the multi-layer perceptron, and *Conv*(·) represents a CNN layer.

### 2.2.3. Convolutional Neural Network

The widely-used CNN in image analysis offers strong grid data processing capabilities [35]. As shown in Figure 4, the input, convolutional, pooling, fully connected, and output layers make up the fundamental architecture of the CNN. The convolution and pooling layers transform and extract features from the information in the input layer. The fully connected layer then performs the mapping between the resulting characteristic maps and output values [16].

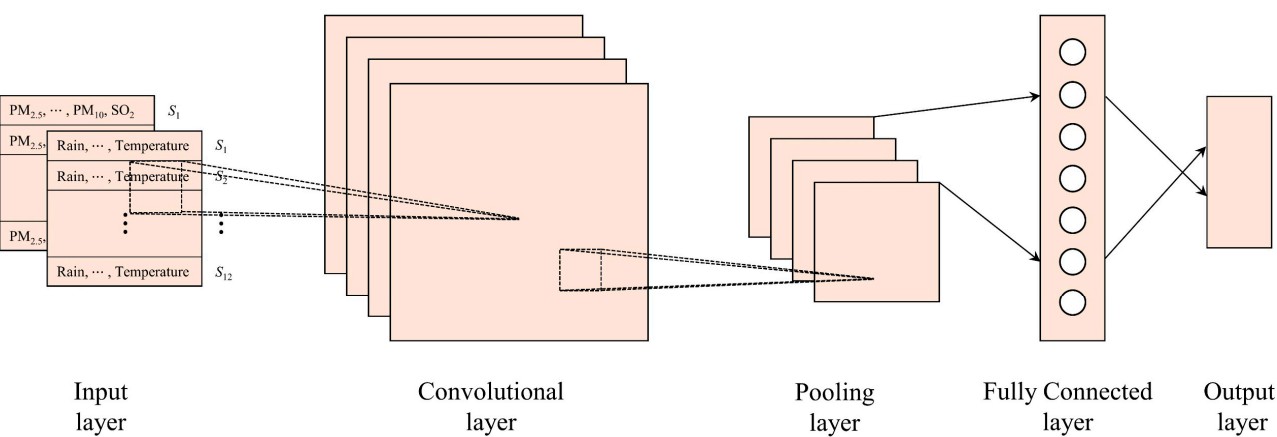

**Figure 4.** The fundamental architecture of CNN.

The convolutional layer, the most important layer in the CNN, extracts the features of the input image by means of the convolutional kernel. The size of the input matrix is larger than the convolution kernel. The convolution layer uses convolution operations to output the characteristic map. Each component of the feature map has the following calculation formula.

$$x_{i,j}^{out} = f_{Conv}\left( \sum_{m=0}^{k} \sum_{n=0}^{k} w_{m,n} x_{i+m,j+n}^{in} + b \right) \tag{6}$$

where $x_{i,j}^{out}$ is the characteristic map's output value for row *i* and column *j*; $x_{i+m,j+m}^{in}$ is the value in the input matrix's row *i* and column *j*; $f_{Conv}(·)$ is activation function; $w_{m,n}$ is the convolution kernel's weight in row *m* and column *n*; and *b* is the convolution kernel's bias [23].

### 2.2.4. Bi-Directional Long Short-Term Memory

In order to overcome the difficulty of the long-term dependence of time series data, the LSTM introduces a special cell storage structure. As shown in Figure 5b, the architecture of

each LSTM cell has an input gate $i_t$, a forgetting gate $f_t$, and an output gate $O_t$. The specific derivation of the LSTM is as follows.

$$f_t = \sigma(W_f \cdot \left[h_{t-1}, x_t\right] + b_f) \tag{7}$$

$$i_t = \sigma(W_i \cdot [h_{t-1}, x_t] + b_i) \tag{8}$$

$$\widetilde{C}_t = \tan h(W_C \cdot [h_{t-1}, x_t] + b_C) \tag{9}$$

$$C_t = f_t * C_{t-1} + i_t * \widetilde{C}_t \tag{10}$$

$$O_t = \sigma(W_o \cdot [h_{t-1}, x_t] + b_o) \tag{11}$$

$$h_t = O_t * \tanh(C_t) \tag{12}$$

where $W_f$, $W_i$, $W_C$, and $W_o$ are the input weights, $b_f$, $b_i$, $b_c$, and $b_o$ are the deviation weights, $t-1$ is the previous time state, $t$ represents the current time state, $x_t$ is the input vector, and $h_t$ represents the output vector. Here, $f_t$ acts as the forget gate and decides what past PM$_{2.5}$ information and other factors should be forgotten from the cell state. $i_t$ expresses the input gate, which makes the decision on what new data to store in the cell state. Lastly, $\widetilde{C}_t$ is a self-recurrent cell in a neuron, and $C_t$ is the LSTM block's internal memory cell.

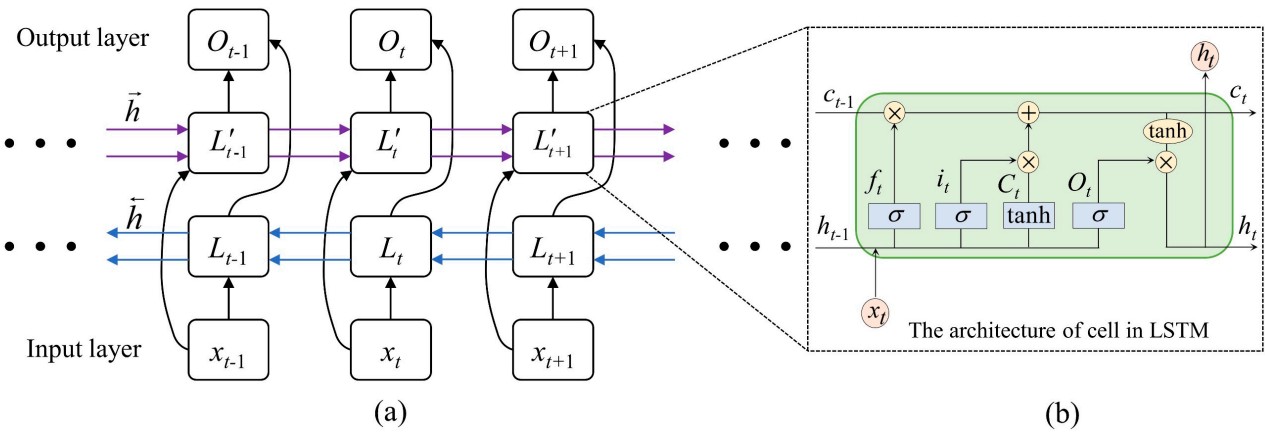

(a)  (b)

**Figure 5.** The basic structure of the Bi LSTM. (**a**) The Bi LSTM. (**b**) The basic structure of LSTM.

The Bi LSTM model, as shown in Figure 5, in contrast to the LSTM model, is made up of a forward LSTM layer $\overrightarrow{h}$ and a backward LSTM layer $\overleftarrow{h}$. Due to the separate hidden layers in both directions, the Bi LSTM can analyze sequence data in both forward and backward directions. Each hidden layer enables the recording of information from the past (forward) and the future (backward) [36]. As a result, a more thorough collection of PM$_{2.5}$ characteristics may be retrieved to raise the network's prediction accuracy.

### 2.3. Experimental Setup

In this experiment, we use Keras based on Tensorflow to construct comparative models of deep learning (CNN, LSTM, Bi LSTM, and CNN-LSTM) and the proposed CBAM-CNN-Bi LSTM model. Table 2 displays the parameters utilized to train the prediction model. Then, we use Adam (adaptive gradient algorithm) as the optimization algorithm and MSE as the loss function for the prediction model. In addition, to enhance the model's capacity for generalization, we packaged the dataset and broke it up, splitting the dataset so that the training set contains 80% of the dataset and the test set contains 20%.

**Table 2.** CBAM-CNN-Bi LSTM model parameters setting.

| Parameters | Value |
|---|---|
| Kernel size of CNN | $3 \times 3$ |
| Convolution channels | 32 |
| Convolution layer | 1 |
| Bi LSTM nodes | 128 |
| Bi LSTM layer | 1 |
| Fully connected layer nodes | 12 |
| Fully connected layer | 1 |
| Learning rate | 0.0001 |
| Batch size | 128 |
| Epochs | 50 |

## 3. Results and Discussion

### 3.1. Performance Evaluation Indices

This paper uses RMSE, MAE, $R^2$, and IA to analyze how well the prediction models performed. The RMSE reflects the sensitivity of the model to error, and the MAE reflects the stability of the model; the closer the value of both to 0, the better the prediction result. $R^2$ represents the ability to forecast the actual data, and IA represents the similarity of the distribution between actual and predicted values, both variables' values span from [0, 1], the closer to 1, the more consistent the predicted results are with the distribution of the true data. The calculation formula is shown as shown.

$$\text{RMSE} = \sqrt{\frac{1}{n}\sum_{i=1}^{n} (y_i - \hat{y}_i)^2} \tag{13}$$

$$\text{MAE} = \frac{1}{n}\sum_{i=1}^{n} |y_i - \hat{y}_i| \tag{14}$$

$$R^2 = \frac{\sum\limits_{i=1}^{n} (\hat{y}_i - \overline{y}_i)^2}{\sum\limits_{i=1}^{n} (y_i - \overline{y}_i)^2} \tag{15}$$

$$\text{IA} = 1 - \frac{\sum_{i=1}^{n} (y_i - \hat{y}_i)^2}{\sum_{i=1}^{n} (|y_i - \overline{y}| + |\hat{y}_i - \overline{y}|)^2} \tag{16}$$

where $n$ denotes the sample size in the dataset, $\hat{y}_i$ denotes predicted value corresponding to it, $y_i$ denotes actual concentration of $PM_{2.5}$, and $\overline{y}$ represents the mean of all measurements of $PM_{2.5}$ concentrations.

### 3.2. Correlation Analysis of Variables

In this subsection, we correlated $PM_{2.5}$ concentrations to achieve two objectives. First, we investigate the relationships between $PM_{2.5}$ concentrations, pollutant concentrations, and meteorological data. Furthermore, to ensure the convergence of the model, we placed the highly correlated factors of influence in the same channel. The correlations between the 12 variables in all sites are shown in Figure 6. For absolute values of correlation, among the pollutants, the highest correlation between $PM_{2.5}$ and itself was found, with $PM_{10}$ (0.89), CO (0.80), $NO_2$ (0.67), $SO_2$ (0.39), and $O_3$ following (−0.17). Among the meteorological factors, wind speed (−0.29) had the strongest association with $PM_{2.5}$, followed by temperature (−0.15) and dew point (0.11). The absolute values of the correlation coefficients for all other influences were below 0.1. In this paper, the six influential factors with the highest absolute values of correlation coefficients with $PM_{2.5}$ are put into the same channel; they are $PM_{2.5}$, $PM_{10}$, CO, $NO_2$, $SO_2$, and wind speed. We put $O_3$ (−0.17), temperature, pressure, dew

point, precipitation, and wind direction into another channel of the model input, forming a 12*6*2 "image".

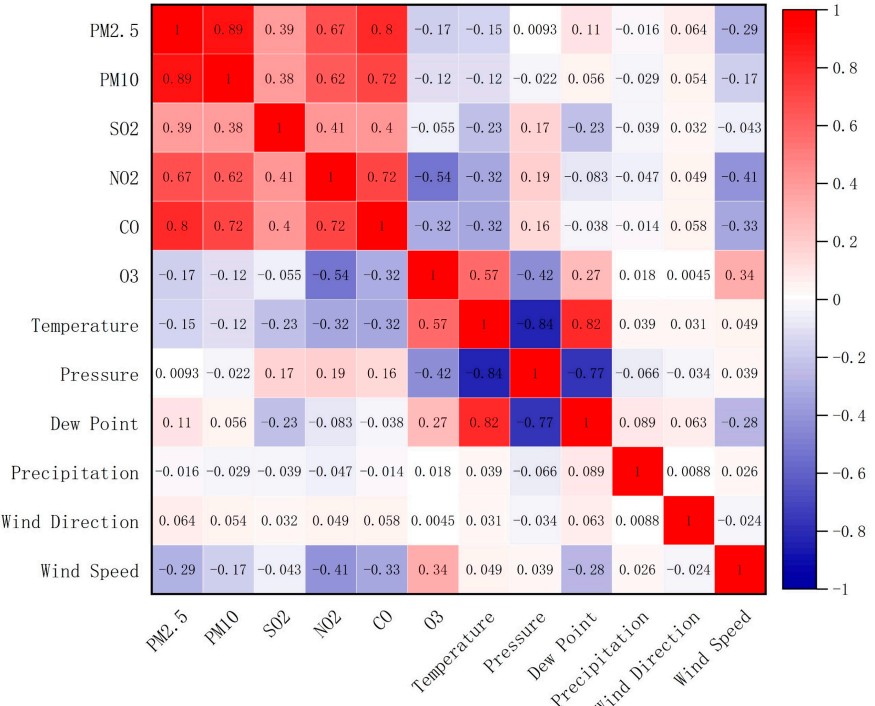

**Figure 6.** Correlation result of multiple variables.

In terms of pollutants, PM$_{2.5}$ concentrations have a highly substantial association with PM$_{10}$, CO, NO$_2$, and SO$_2$, and a bad relationship with O$_3$ concentrations. This is due to the fact that PM$_{2.5}$, PM$_{10}$, CO, NO$_2$, and SO$_2$ are mostly derived from human activities such as coal combustion, vehicle exhausts, and industrial manufacturing. In addition, emissions of PM$_{10}$, CO, NO$_2$, and SO$_2$ contribute to a certain extent to the increase in PM$_{2.5}$ concentrations. Unlike other pollutants, O$_3$ comes mainly from nature. As light and temperature increase, the concentration of O$_3$ increases. Due to the high concentration of O$_3$, there is a chance that some photochemical reactions will consume some PM$_{2.5}$ and lower its concentration.

Compared to pollutants, meteorological factors have a relatively small but integral impact on PM$_{2.5}$ concentrations. The relationship between wind speed and PM$_{2.5}$ concentrations is negative. High wind speeds are conducive to PM$_{2.5}$ dispersion and therefore have a substantial impact on the concentration of PM$_{2.5}$. The increase in temperature causes instability in the atmosphere, which has a positive effect on the dispersion of PM$_{2.5}$. The dew point, a measure of air humidity, is positively correlated with PM$_{2.5}$ concentrations. An environment with high air humidity contributes to the formation of fine particulate matter, making PM$_{2.5}$ less likely to disperse. Wind direction, barometric pressure, and rainfall are all weakly correlated with PM$_{2.5}$ and their changes have little effect on PM$_{2.5}$ concentrations.

### 3.3. Short-Term Prediction

Pollutant prediction models can be divided into short-term prediction models and long-term prediction models [37]. Short-term prediction focuses on the accuracy of the forecast and ensures the safety of human activities in the short term by keeping the forecast time within 12 h [38]. We will compare and analyze the short-term prediction performance of each model for pollutant concentrations in this section.

### 3.3.1. Effect of Convolution Kernel Size on Experimental Results

The convolutional kernel is a key part of the convolutional neural network model, which directly affects how well the features are extracted and how quickly the network converges. The size of the convolution kernel should be appropriate to the size of the input "image". If the convolution kernel is too large, the local features cannot be extracted effectively; if the convolution kernel is too small, the overall features cannot be extracted successfully. Therefore, a convolution kernel of the right size should be selected to fit the input "image". As the size of the input "image" of the CBAM-CNN-Bi LSTM model is 12*6*2, we set the convolution kernel to 2*2, 3*3, 4*4, 5*5 to forecast the $PM_{2.5}$ concentrations in the next 6 h.

Table 3 gives the average of the performance evaluation indicators of the CBAM-CNN-Bi LSTM for $PM_{2.5}$ concentration prediction for the next 6 h at different convolutional kernel sizes. As shown in Table 3, the test errors of the models do not differ significantly when the size of the convolution kernel varies. However, when the convolutional kernel size was 3*3, RMSE and MAE reached a minimum value of 29.65 and 18.58, and $R^2$ and IA reached a maximum value of 0.8192 and 96.01%.

**Table 3.** Effect of convolution kernel size on experimental results in CBAM-CNN-Bi LSTM. (RMSE is the root mean squared error, MAE is the mean squared error, $R^2$ is the r- squared, IA is the index of agreement).

| Convolution Kernel Size | RMSE | MAE | $R^2$ | IA |
|:---:|:---:|:---:|:---:|:---:|
| 2*2 | 29.79 | 18.84 | 0.8153 | 95.92% |
| 3*3 | 29.65 | 18.58 | 0.8192 | 96.01% |
| 4*4 | 30.26 | 19.12 | 0.8141 | 95.82% |
| 5*5 | 30.14 | 18.83 | 0.8162 | 95.85% |

Note: window size = 6 (the model's input window size represents historical observations), and model performance evaluation indicators (RMSE, MAE, $R^2$ and IA) are the average of prediction of the next 1–6 h.

This demonstrates that the highest prediction performance for our proposed model occurs when the convolution kernel is 3*3. When the convolution kernel is smaller than 3*3, the model has an underfitting problem for the overall spatial features of $PM_{2.5}$ concentrations; when the size of the kernel is larger than 3*3, our model cannot effectively extract the local spatial features of $PM_{2.5}$ concentrations. When the convolution kernel is 3*3, the model is capable of extracting both global and local spatial information related to $PM_{2.5}$ concentrations. In the following experiments, the convolution kernel size is set to 3*3.

### 3.3.2. Effect of Different Models on Experimental Results

The quantitative results for single-step $PM_{2.5}$ concentrations prediction are given in Table 4, which gives a comparison of the RMSE, MAE, $R^2$ and IA for CNN, LSTM, Bi LSTM, CNN-LSTM and CBAM-CNN-Bi LSTM. As shown in Table 4, our proposed model performed better than other deep learning models in single-step $PM_{2.5}$ concentration prediction. In contrast to other models, our proposed model has the minimum prediction error and the greatest prediction accuracy and reduces RMSE to 18.90, MAE to 11.20, improves $R^2$ to 0.9397, and IA to 98.54%. However, the prediction results of our proposed model are not much ahead of Bi LSTM because the single-step prediction is relatively simple and does not reflect the advantages of our designed architecture. In addition, CNN has the worst prediction performance, and LSTM and Bi LSTM perform predictions more accurately than CNN-LSTM. This means that the LSTM and Bi LSTM, with their excellent time series data processing capability, are more suitable than CNN-LSTM for the $PM_{2.5}$ concentration single-step prediction task.

**Table 4.** Performance evaluation indicators for model single-step prediction.

| Models | RMSE | MAE | $R^2$ | IA |
|--------|------|-----|-------|-----|
| CNN | 23.15 | 14.15 | 0.9136 | 97.83% |
| LSTM | 19.53 | 11.49 | 0.9314 | 98.39% |
| Bi LSTM | 18.94 | 11.35 | 0.9370 | 98.30% |
| CNN-LSTM | 20.51 | 11.99 | 0.9313 | 98.31% |
| CBAM-CNN-Bi LSTM | 18.90 | 11.20 | 0.9397 | 98.54% |

Note: window size = 3, and model performance evaluation indicators (RMSE, MAE, $R^2$ and IA) are the prediction of the next 1 h.

It is well known that with the increase in forecast time steps, forecasting becomes more difficult. To further evaluate the $PM_{2.5}$ concentrations short-term prediction capability of CBAM-CNN-Bi LSTM and other deep learning models, we predicted $PM_{2.5}$ concentrations for the next 2–12 h and presented the predicted quantitative results through the change curves of RMSE, MAE, $R^2$ and IA in Figure 7. As shown in Figure 7, the predictive ability of all prediction models declines as the prediction time step rises. We observed that four performance evaluation indicators of CNN were always worse than other deep learning models, and the LSTM and Bi LSTM prediction performance was generally consistent. Interestingly, as shown in Figure 7a,b, in cases where the predicted time step is under five, compared to the LSTM and Bi LSTM, the CNN-LSTM has a greater prediction error. Does this mean that CNN-LSTM has poorer short-term prediction performance than LSTM and Bi LSTM? In fact, we will find that four metrics for evaluating the performance of the CNN-LSTM start to outperform the LSTM and Bi LSTM when the prediction step size is greater than five. This suggests that CNN-LSTM, with its hybrid model structure, can better quantify complex data when prediction problems become difficult. It is worth noting that when prediction time steps are greater than 4, our proposed model consistently maintains optimal prediction performance with the lowest RMSE and MAE, and the highest $R^2$ and IA.

In summary, for $PM_{2.5}$ concentrations short-term prediction, when the convolution kernel is 3*3, CBAM-CNN-Bi LSTM obtains the best prediction performance and maintains the best results among all deep learning models. This means that CBAM plays a key role in the prediction of deep learning models. CBAM obtains the feature relationship between pollutant data and meteorological data, optimizes the CNN spatial feature extraction, and improves the model prediction accuracy.

*3.4. Long-Term Prediction*

The research on pollutant concentration prediction has mainly focused on pollutant concentration short-term prediction, but this is not sufficient to meet the actual demand. The purpose of long-term forecasting is to forecast pollutant concentrations for a longer period of time in the future, and its predictions can serve as a useful reference for managers. It can be seen that long-term predictions of pollutant concentrations are very meaningful. We will analyze the pollutant concentration long-term prediction performance of each model in this section.

3.4.1. $PM_{2.5}$ Concentration Prediction

The quantitative results of the long-term $PM_{2.5}$ concentration prediction (h13~h18) are given in Table 5, which gives the comparison of RMSE, MAE, $R^2$, and IA for CNN, LSTM, Bi LSTM, CNN-LSTM, and our proposed model. As shown in Table 5, our proposed model outperforms other deep learning models in $PM_{2.5}$ concentration prediction (h13~h18). In comparison to alternative models, CBAM-CNN-Bi LSTM has the lowest prediction error and the highest prediction accuracy and reduces RMSE to 37.33, MAE to 26.54, $R^2$ to 0.6981, and IA to 93.50%. As shown in Table 5, the $R^2$ of CNN has decreased to −0.2546, which indicates that CNN is inappropriate for long-term $PM_{2.5}$ concentration prediction. It is worth noting that the prediction performance of CNN-LSTM is superior to Bi LSTM and

LSTM, this shows that the hybrid model of CNN-LSTM is more suitable for long-term PM$_{2.5}$ concentration prediction.

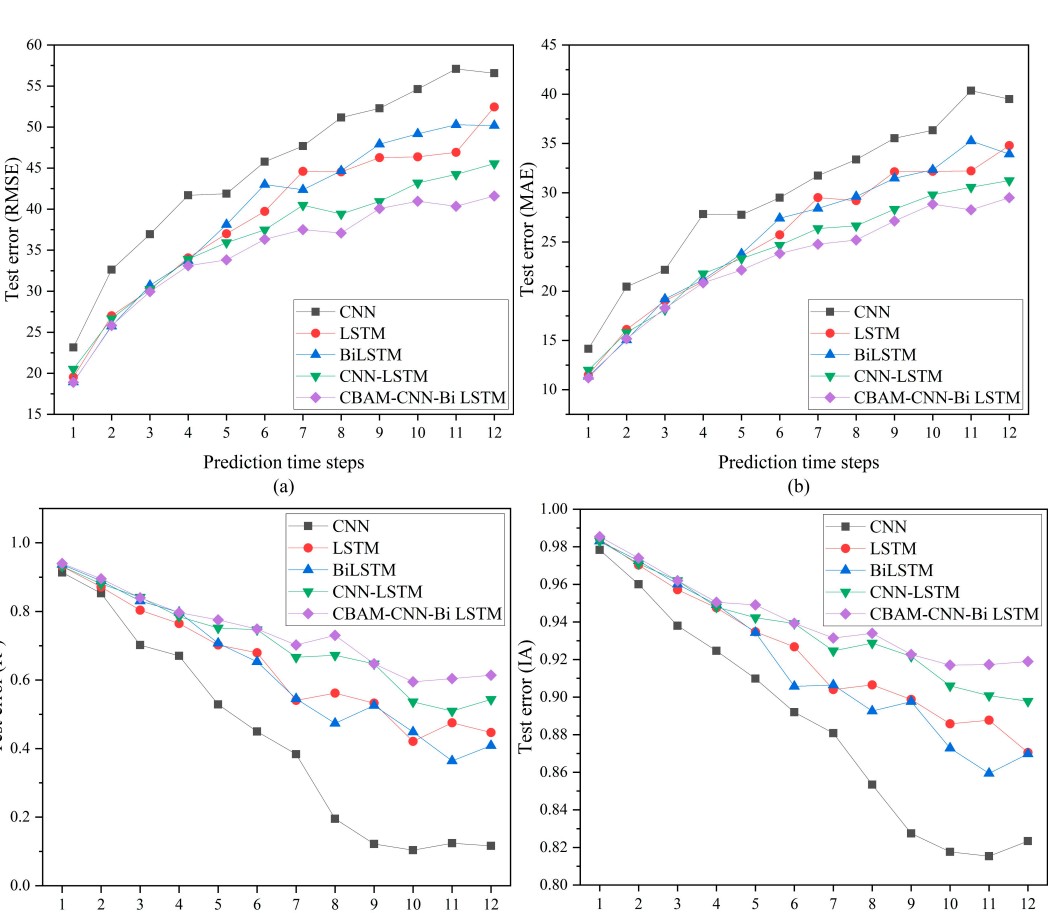

**Figure 7.** RMSE, MAE, R$^2$, and IA of the CBAM-CNN-Bi LSTM model at different prediction time steps and comparisons with other deep learning models. (**a**) RMSE of models at different prediction steps. (**b**) MAE of models at different prediction steps. (**c**) R$^2$ of models at different prediction steps. (**d**) IA of models at different prediction steps.

**Table 5.** Performance evaluation indicators for model prediction (h13–h18).

| Models | RMSE | MAE | R$^2$ | IA |
|---|---|---|---|---|
| CNN | 58.79 | 41.53 | −0.2546 | 79.05% |
| LSTM | 49.71 | 34.87 | 0.3561 | 87.03% |
| Bi LSTM | 48.84 | 34.89 | 0.4132 | 87.73% |
| CNN-LSTM | 42.84 | 30.41 | 0.5876 | 91.12% |
| CBAM-CNN-Bi LSTM | 37.33 | 26.54 | 0.6981 | 93.50% |

Note: window size = 18, and model performance evaluation indicators (RMSE, MAE, R$^2$, and IA) are the average of prediction of the next 13–18 h.

Next, we analyze the effect of the size of the prediction time steps on CNN, LSTM, Bi LSTM, CNN-LSTM, and our proposed model. As shown in Tables 6–8, when the prediction time steps are the same as h13~h18, the larger the window size, the more accurate the model's prediction performance. This means that deep learning models can optimize the prediction results by learning more historical data. Table 5 shows that the prediction performance of all deep learning methods steadily declines as prediction step size increases. It is evident that, in contrast to other deep learning methods, our proposed model has the

minimum prediction error (RMSE and MAE) and the highest prediction accuracy ($R^2$ and IA) for different prediction time steps.

**Table 6.** Performance evaluation indicators for model prediction (h13–h24).

| Models | RMSE | | MAE | | $R^2$ | | IA | |
|---|---|---|---|---|---|---|---|---|
| | 13–18 h | 19–24 h | 13–18 h | 19–24 h | 13–18 h | 19–24 h | 13–18 h | 19–24 h |
| CNN | 58.53 | 62.59 | 42.28 | 45.33 | −0.3338 | −1.0071 | 78.25% | 72.26% |
| LSTM | 47.99 | 52.94 | 34.54 | 37.73 | 0.4216 | 0.2746 | 88.06% | 85.25% |
| Bi LSTM | 45.42 | 49.45 | 32.72 | 35.57 | 0.5182 | 0.3972 | 89.67% | 87.44% |
| CNN-LSTM | 40.30 | 43.44 | 28.54 | 31.45 | 0.6386 | 0.5525 | 92.20% | 90.57% |
| CBAM-CNN-Bi LSTM | 34.75 | 37.00 | 24.70 | 26.53 | 0.7508 | 0.6942 | 94.49% | 93.49% |

Note: window size = 24, and model performance evaluation indicators (RMSE, MAE, $R^2$ and IA) are the average of prediction of the next $t \sim t + n$ hours.

**Table 7.** Testing error for model prediction (h13–h48).

| Models | RMSE | | | MAE | | |
|---|---|---|---|---|---|---|
| | 13–18 h | 19–24 h | 25–48 h | 13–18 h | 19–24 h | 25–48 h |
| CNN | 57.19 | 61.82 | 65.69 | 40.41 | 44.63 | 48.19 |
| LSTM | 46.03 | 48.60 | 49.34 | 32.76 | 35.49 | 35.90 |
| Bi LSTM | 43.07 | 44.94 | 45.68 | 31.05 | 32.03 | 33.43 |
| CNN-LSTM | 37.20 | 38.55 | 40.24 | 26.51 | 27.45 | 29.16 |
| CBAM-CNN-Bi LSTM | 31.47 | 31.84 | 32.34 | 21.86 | 21.78 | 22.30 |

Note: window size = 48, and prediction error is averaged out by model testing error (RMSE and MAE). of the future $t \sim t + n$ hours.

**Table 8.** Testing accuracy for model prediction (h13–h48).

| Models | $R^2$ | | | IA | | |
|---|---|---|---|---|---|---|
| | 13–18 h | 19–24 h | 25–48 h | 13–18 h | 25–36 h | 25–48 h |
| CNN | −0.1841 | −0.9108 | −1.3622 | 80.01% | 73.68% | 68.79% |
| LSTM | 0.5019 | 0.3845 | 0.3467 | 89.44% | 87.53% | 87.02% |
| Bi LSTM | 0.5538 | 0.5294 | 0.4930 | 90.76% | 90.03% | 89.50% |
| CNN-LSTM | 0.7190 | 0.7077 | 0.6637 | 93.74% | 93.31% | 92.52% |
| CBAM-CNN-Bi LSTM | 0.8068 | 0.8042 | 0.7980 | 95.52% | 95.50% | 95.39% |

Note: window size = 48, and model testing accuracy ($R^2$ and IA) are the average of prediction accuracy of the next $t \sim t + n$ hours.

To verify the effectiveness of our proposed model, we analyzed the variations of RMSE, MAE, $R^2$, and IA for each model within the prediction step of 48 h. As shown in Tables 7 and 8, the four performance evaluation indicator metrics of CNN, Bi LSTM, LSTM, and CNN-LSTM fluctuated widely in the long-term prediction. Furthermore, what is interesting about the data in Tables 7 and 8 is that our proposed model can continue to outperform the CNN-LSTM significantly as the prediction time step grows. In addition, the four evaluation indicators of our proposed model fluctuated less (with little change in values) as the prediction time steps increased, which shows that CBAM-CNN-Bi LSTM is most suitable for long-term PM$_{2.5}$ concentration prediction. In conclusion, our proposed model can be used to maintain the best and most stable prediction performance in long-term PM$_{2.5}$ concentration prediction.

To further validate the prediction performance of our proposed model, we analyzed the fitting ability of our proposed model and four other deep learning models for PM$_{2.5}$ concentrations at a prediction time step of 48 h. As shown in Figure 8a1–e1, we found that the CNN has the worst prediction performance and cannot describe the trend of PM$_{2.5}$ concentrations. Compared to LSTM and Bi LSTM, CNN-LSTM has a higher long-term prediction ability for PM$_{2.5}$ concentrations, but the accuracy of the prediction of sudden

change points of PM$_{2.5}$ concentrations is not enough. Our proposed model shown in Figure 8 outperforms other comparative models in the prediction of sudden change points in PM$_{2.5}$ concentrations. We observed that when the PM$_{2.5}$ concentrations were larger than 200 μg/m$^3$, the comparison model's predicted outcomes were unable to capture the true trend. This also reflects that when the PM$_{2.5}$ concentrations are too high, it makes precise prediction using the model challenging. Moreover, the predictions of our proposed model and the observed outcomes are virtually identical (as shown in the red wireframe part in Figure 8). This means that our proposed model has a good fit for the prediction of high PM$_{2.5}$ concentration values.

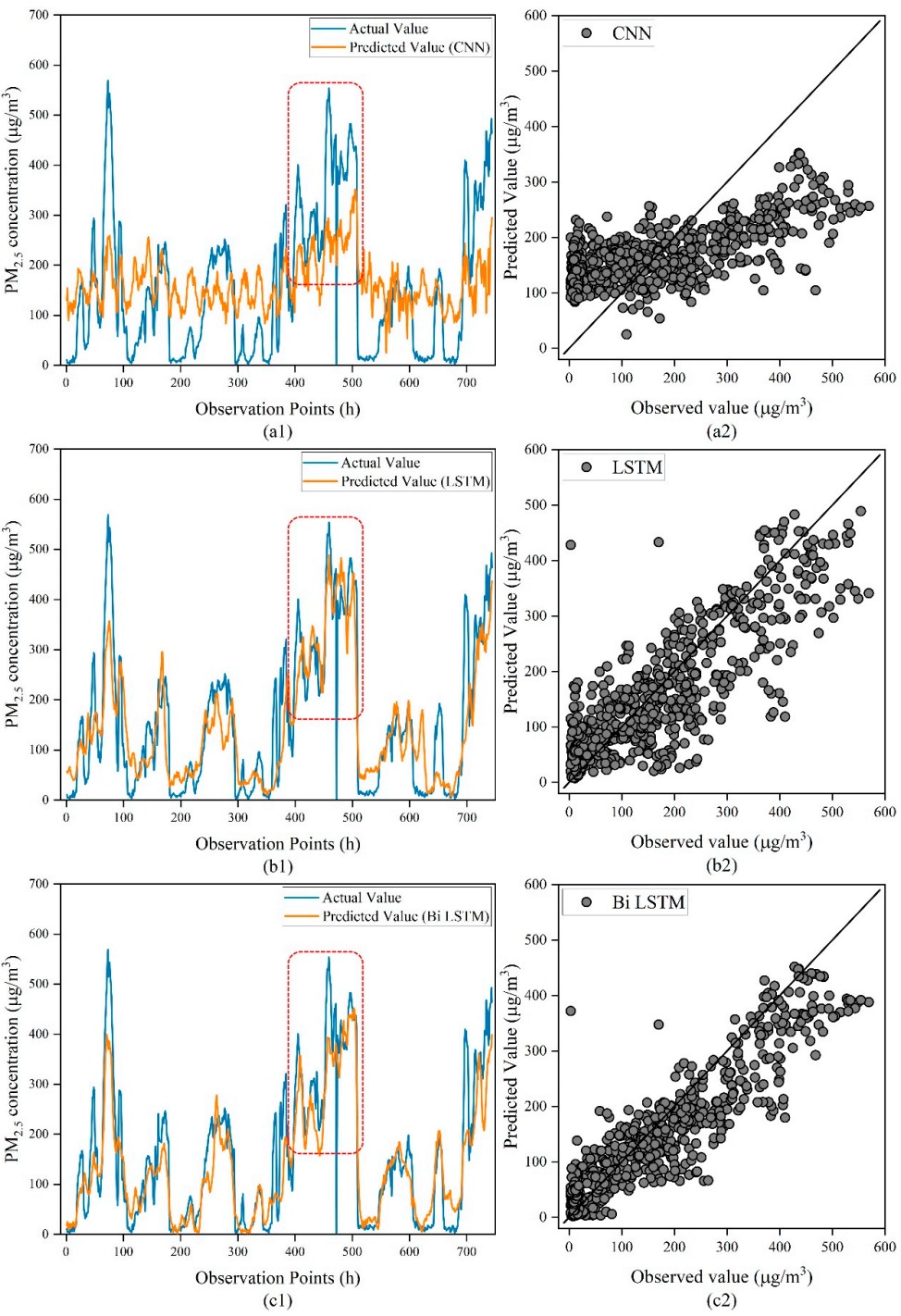

**Figure 8.** *Cont.*

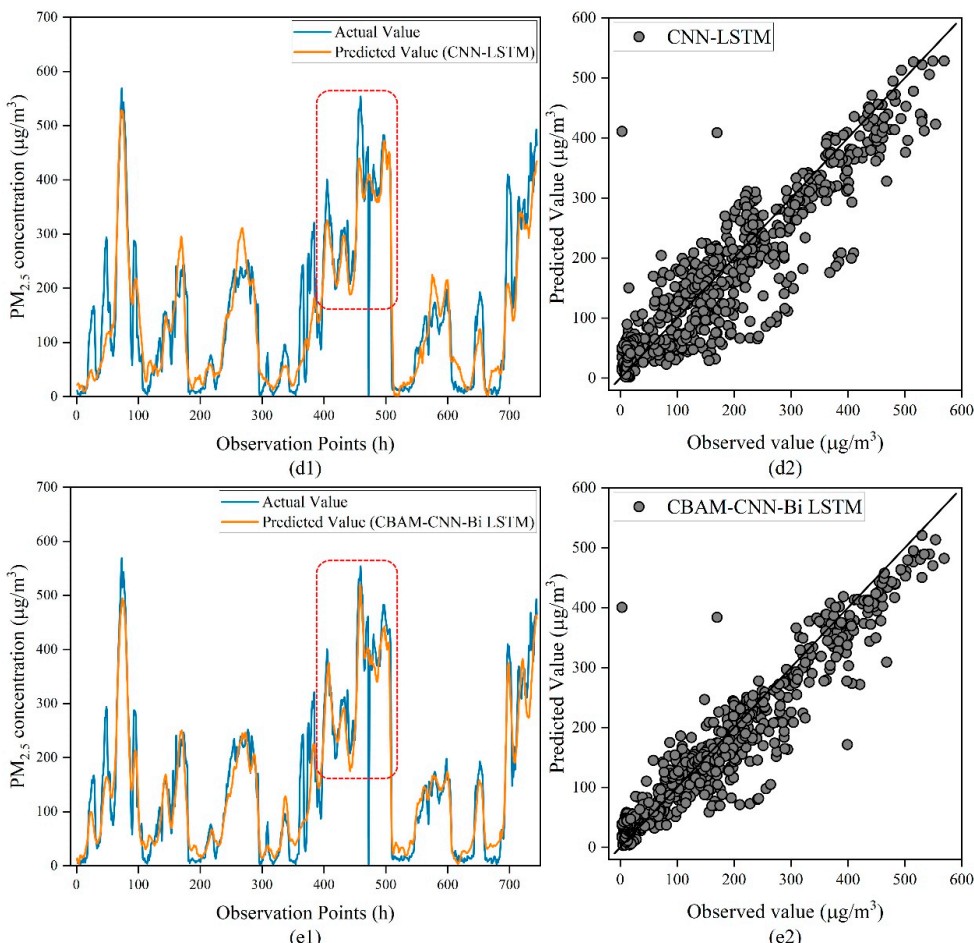

**Figure 8.** Comparison of PM$_{2.5}$ concentration prediction models for the next 48h at station 1001A. (**a1**) Prediction line graph of CNN model; (**a2**) Prediction scatter plot of CNN model; (**b1**) Prediction line graph of LSTM model; (**b2**) Prediction scatter plot of LSTM model; (**c1**) Prediction line graph of Bi LSTM model; (**c2**) Prediction scatter plot of Bi LSTM model; (**d1**) Prediction line graph of CNN-LSTM model; (**d2**) Prediction scatter plot of CNN-LSTM model; (**e1**) Prediction line graph of CBAM-CNN-Bi LSTM model; (**e2**) Prediction scatter plot of CBAM-CNN-Bi LSTM model.

Combined with the capability of fitting the model in Figure 8, we can draw the following conclusions: (1) CBAM-CNN-Bi LSTM has better prediction performance in all time periods and can effectively forecast PM$_{2.5}$ concentrations in different environments; (2) in the case of high PM$_{2.5}$ concentration values, the CBAM-CNN-Bi LSTM has a good fitting effect, making the predicted and observed values basically consistent; (3) we can see that the number of sudden change point samples is relatively small. This phenomenon causes the deep learning model to insufficiently learn the change pattern of PM$_{2.5}$ concentrations in the case of sudden changes. This is the reason why the deep learning model is difficult to fit to the sudden change points.

### 3.4.2. Other Pollutant Concentration Prediction

To confirm the applicability of our suggested model. In this section, PM$_{10}$ and SO$_2$ are used as examples, and Tables 9 and 10 display the evaluation indices of model prediction performance. As shown in Table 9, our proposed model still has the best prediction ability. The prediction performance of our proposed model is significantly better than that of the single framework model. Compared with the CNN-LSTM model, our proposed model reduces RMSE by 3.57, MAE by 2.65, R$^2$ by 0.0115, and IA by 0.59%. This indicates that the convolutional block attention module optimizes the model and improves the prediction accuracy. The predicted results for SO$_2$ are similar to those for PM$_{10}$. As shown in Table 10,

our model reduced the RMSE to 9.87, MAE to 6.14, $R^2$ to 0.7175, and IA to 93.76%. In the prediction of $SO_2$ and $PM_{10}$, our model also maintains the optimal prediction results. This indicates that our proposed prediction approach is applicable to the prediction of other pollutants and is as successful.

**Table 9.** Performance evaluation indicators for model prediction of $PM_{10}$ (h13–h48).

| Models | RMSE | MAE | $R^2$ | IA |
|---|---|---|---|---|
| CNN | 73.05 | 53.15 | −1.1639 | 70.69% |
| LSTM | 56.12 | 41.12 | 0.3816 | 87.19% |
| Bi LSTM | 54.97 | 40.77 | 0.4212 | 88.09% |
| CNN-LSTM | 45.92 | 32.72 | 0.6818 | 92.74% |
| CBAM-CNN-Bi LSTM | 42.35 | 30.07 | 0.6933 | 93.33% |

Note: window size = 48, and model performance evaluation indicators (RMSE, MAE, $R^2$ and IA) are the average of prediction of the next 13–48 h.

**Table 10.** Performance evaluation indicators for the model prediction of $SO_2$ (h13–h48).

| Models | RMSE | MAE | $R^2$ | IA |
|---|---|---|---|---|
| CNN | 15.19 | 9.59 | −0.0793 | 80.79% |
| LSTM | 13.54 | 8.38 | 0.3172 | 86.35% |
| Bi LSTM | 13.15 | 8.22 | 0.3445 | 87.11% |
| CNN-LSTM | 11.23 | 7.05 | 0.6568 | 92.53% |
| CBAM-CNN-Bi LSTM | 9.87 | 6.14 | 0.7175 | 93.76% |

Note: window size = 48, and model performance evaluation indicators (RMSE, MAE, $R^2$, and IA) are the average of prediction of the next 13–48 h.

## 4. Conclusions and Future

In my research, a unique $PM_{2.5}$ concentration prediction model (CBAM-CNN-Bi LSTM) is proposed, which gives a reasonable prediction by learning from a large amount of pollutant data and meteorological data. CBAM-CNN-Bi LSTM is a hybrid deep learning model which consists of CBAM, CNN, and Bi LSTM. The advantages of CBAM-CNN-Bi LSTM are concluded as below:

(1) By utilizing the convolutional block attention module, the CNN network degradation issue may be solved. The spatial attention module assists CNN in efficiently acquiring spatial correlation features between multiple sites to extract pollutants and meteorological data. The channel attention module is used to capture the complex relationship features between the influencing factors of model inputs. Convolutional block attention modules optimize convolutional neural networks to provide more reliable data for more precise result prediction;

(2) By using Bi LSTM as the output prediction layer, the model not only obtains the performance advantage of long-time series prediction through Bi LSTM, avoiding the problem of underutilization of contextual information, but also extracts the effective association features of the output of the convolutional neural network layer to achieve the goal of mining data spatiotemporal association;

(3) Our proposed model can be simultaneously applied to meteorological and pollution data from multiple stations for environmental monitoring of big data while considering the changes in the spatial and temporal distribution of the data to achieve the prediction of air pollutant concentrations in the target city. Experiments conducted on the dataset show that our framework obtains better results than other methods.

Based on the aforementioned experimental findings, the effectiveness of our proposed model is demonstrated. In comparison to other models, our proposed model gives accurate $PM_{2.5}$ concentration predictions by fully extracting the temporal and spatial characteristics of $PM_{2.5}$ and the correlation between pollutant data and meteorological data and overcoming the problem of long-time dependence of $PM_{2.5}$ concentrations. Therefore, our proposed

model overcomes the weaknesses of CNN-LSTM and has more practical value. However, traffic, vegetation cover, and pedestrian flow are not considered in this paper, which will be addressed in future studies.

**Author Contributions:** Conceptualization, D.L.; methodology, D.L.; software, D.L.; validation, D.L.; formal analysis, Y.Z.; investigation, J.L.; resources, J.L.; data curation, D.L.; writing—original draft preparation, D.L.; writing—review and editing, D.L. and Y.Z.; visualization, Y.Z.; supervision, J.L.; project administration, Y.Z.; funding acquisition, Y.Z. All authors have read and agreed to the published version of the manuscript.

**Funding:** This work was supported by Lanzhou Jiaotong University (grant no. EP 201806).

**Institutional Review Board Statement:** Not applicable.

**Informed Consent Statement:** Not applicable.

**Data Availability Statement:** Restrictions apply to the availability of these data. Data was obtained from [Songxi Chen] and are available [https://archive.ics.uci.edu/ml/datasets/Beijing+Multi-Site+Air-Quality+Data] with the permission of [Songxi Chen].

**Conflicts of Interest:** The authors declare no conflict of interest.

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
