# Peer review of "Prediction of Multi-Site PM2.5 Concentrations in Beijing Using CNN-Bi LSTM with CBAM"

_atmosphere, doi:10.3390/atmos13101719_

Round 1

Reviewer 1 Report

Air pollution is a growing problem and challenges people's healthy lives. The results show that in comparison to other models, CBAM-CNN-Bi LSTM improves the accuracy of PM2.5 concentration prediction. Our proposed prediction model performs well for the prediction tasks from 1 to 12 hours. For the 13 to 48 hours prediction task, the CBAM-CNN-Bi LSTM also achieves satisfactory results. The study needs the following changes:

i) Add possible research questions in the introduction and link them to the studied objectives.

ii) Further, discussed the novelty of the study with research hypotheses.

iii) Results should be critically discussed and linked to earlier studies.

iv) Avoid too many abbreviations in the study, and

v) write short-term, medium-term, and long-term policy implications. 

Author Response

Point 1. Add possible research questions in the introduction and link them to the studied objectives.

Response 1: Thank you for your suggestion. We agree with the reviewer.

According to the suggestions of the reviewer, we have sorted out previous related work on pollutant prediction and other aspects in the revised manuscript, raised relevant questions and objectives, and revised the introductory part of the paper.

Please refer to the introduction part of the revised manuscript.

Point 2. Further, discussed the novelty of the study with research hypotheses.

Response 2: Thank you for your suggestion. We agree with the reviewer.

According to the suggestion of the reviewer, we have added the novelty and research hypothesis of the CBAM-CNN-Bi LSTM model proposed in this paper in the revised manuscript.

Please refer to the introduction part of the revised manuscript.

Point 3. Results should be critically discussed and linked to earlier studies.

Response 4: Thank you for your suggestion. We agree with the reviewer.

According to the suggestion of the reviewer, we have made some changes to the experimental part of Part IV of the article.

Please refer to the revised draft.

Point 4. Avoid too many abbreviations in the study.

Response 4: Thank you for your suggestion. We agree with the reviewer.

According to the suggestion of the reviewer, we have revised some of the abbreviations in the revised version.

Please refer to the revised draft.

Point 5. Write short-term, medium-term, and long-term policy implications.

Response 5: Thank you for your suggestion. We agree with the reviewer.

According to the suggestion of the reviewer, we have added the novelty and research hypothesis of the CBAM-CNN-Bi LSTM model proposed in this paper in the revised manuscript.

Please refer to the Section3.3 and Section 3.4 part of the revised manuscript, and the specific revisions are as follows:

Due to time constraints, only the description of language errors in the article will be corrected. We will touch up the article in all aspects next time if necessary.

Reviewer 2 Report

This paper titled “Prediction of Multi-site PM2.5 Concentrations in Beijing Using CNN-Bi LSTM with CBAM” presents an interesting proposal. However, some issues should be addressed to improve the quality of the paper. After further revision, this paper can be accepted for publication.

1. The difference between the previous studies and your work is not clear. What is the research gap? How do you bridge the gap?

2. The main work and contribution of your study should be summarized in the introduction.

3. Please add the details about the organization of this study in the introduction.

4. To further validate the effectiveness of the proposed method, additional case study from other widely used city should be added as a benchmark.

5. The model only based on PM2.5 concentrations and the Random Walk (persistence) model should be added as benchmark.

6. What is the basis for determining the ratio of training and testing? Is the amount of data is too large? Can the data volume of the dataset be decreased? At the same time, can all the PM2.5 concentrations data series information of different sites be fully displayed in a figure?

7. Is it possible to share the script to replicate the results?

Author Response

Point 1. The difference between the previous studies and your work is not clear. What is the research gap? How do you bridge the gap?

Response 1: Thank you for your suggestion. We agree with the reviewer.

According to the suggestions of the reviewer, we have sorted out previous related work on pollutant prediction and other aspects in the revised manuscript, raised relevant questions and solutions, and revised the introductory part of the paper.

Please refer to the introduction part of the revised manuscript.

Point 2. The main work and contribution of your study should be summarized in the introduction.

Response 2: Thank you for your suggestion. We agree with the reviewer.

According to the suggestions of the reviewer, we have added the main work and contributions of our study to the revised manuscript.

Please refer to the introduction part of the revised manuscript.

Point 3. Please add the details about the organization of this study in the introduction.

Response 3: Thank you for your suggestion. We agree with the reviewer.

According to the suggestions of the reviewer, we have added the organization of our study to the revised draft.

Please refer to the introduction part of the revised manuscript.

Point 4. To further validate the effectiveness of the proposed method, additional case study from other widely used city should be added as a benchmark.

Response 4: Thank you for your suggestion. We agree with the reviewer.

We would also like to add a few more cities as study areas, but currently only the Beijing dataset is available. Data from other cities are being collected and we will add experimental areas later. I strongly agree with your suggestion, but I am very sorry.

Point 5. The model only based on PM2.5 concentrations and the Random Walk (persistence) model should be added as benchmark.

Response 5: Thank you for your suggestion. I am sorry, please forgive me for not understanding what you mean by this suggestion. As I understood it means adding other pollution concentration predictions to prove the usefulness of the model. We combed through many papers on related research and did not find any literature related to the Random Walk (persistence) model. We will do our best to match your suggestions if needed for follow-up work.

Please refer to the section 3.4.2 of the revised manuscript.

Point 6. What is the basis for determining the ratio of training and testing? Is the amount of data is too large? Can the data volume of the dataset be decreased? At the same time, can all the PM2.5 concentrations data series information of different sites be fully displayed in a figure?

Response 6: Thank you for your suggestion. In the task of deep learning, the common ratio of training data to test data is divided into 7:3, 8:2, and 9:1. After experiments, it is found that when the ratio of training data to test data is 8:2, the results of test data are optimal, so 8:2 is used.

Since the depth of deep learning models is generally deep and the number of parameters is large, it is easy to overfit if the amount of data is small.

We present the PM2.5 concentration data series information for different locations on Figure 1(b).

Please refer to the section 2.1.1 of the revised manuscript.

Point 7. Is it possible to share the script to replicate the results?

Response 7 : The system can only submit word or pdf files and we can only copy the code to word. Since the article is not yet published, I can only send you the CBAM code. Please excuse me for this. If the article is accepted, I will post the code on the github URL.

Round 2

Reviewer 2 Report

Since this paper is well revised, it is recommended to be accepted.